# Saliva as a useful tool for evaluating upper mucosal antibody response to influenza

**Yasuko Tsunetsugu-Yokota**[1,2]*, **Sayaka Ito**[1,3], **Yu Adachi**[2], **Taishi Onodera**[2], **Tsutomu Kageyama**[4], **Yoshimasa Takahashi**[2]

1 Department of Medical Technology, School of Human Sciences, Tokyo University of Technology, Tokyo, Japan, 2 Research Center for Drug and Vaccine Development, National Institute of Infectious Diseases, Tokyo, Japan, 3 Department of Health Sciences, Saitama Prefectural University, Saitama, Japan, 4 Center for Emergency Preparedness and Response, National Institute of Infectious Diseases, Tokyo, Japan

* yokotaysk@stf.teu.ac.jp

## Abstract

Mucosal immunity plays a crucial role in controlling upper respiratory infections, including influenza. We established a quantitative ELISA to measure the amount of influenza virus-specific salivery IgA (sIgA) and salivary IgG (sIgG) antibodies using a standard antibody broadly reactive to the influenza A virus. We then analyzed saliva and serum samples from seven individuals infected with the A(H1N1)pdm09 influenza virus during the 2019–2020 flu seasons. We detected an early (6–10 days post-infection) increase of sIgA in five of the seven samples and a later (3–5 weeks) increase of sIgG in six of the seven saliva samples. Although the conventional parenteral influenza vaccine did not induce IgA production in saliva, vaccinated individuals with a history of influenza infection had higher basal levels of sIgA than those without a history. Interestingly, we observed sIgA and sIgG in an asymptomatic individual who had close contact with two influenza cases. Both early mucosal sIgA secretion and late systemically induced sIgG in the mucosal surface may protect against virus infection. Despite the small sample size, our results indicate that the saliva test system can be useful for analyzing upper mucosal immunity in influenza.

## Introduction

Influenza A virus (IAV) causes an acute upper respiratory disease in humans, resulting in seasonal influenza and occasional severe lung diseases of zoonotic origin [1]. Because of the ever-mutating IAV genome, the IAV vaccine is updated every year based on global surveillance and the clinical data on the circulating virus strains during the winter season in the opposite hemisphere [2]. Still, influenza causes almost 650,000 deaths worldwide every year [3].

The situation with influenza has dramatically changed because of the emergence of a novel coronavirus, SARS-CoV-2, in late 2019. The COVID-19 pandemic, caused by SARS-CoV-2 infection, has overwhelmed influenza-related fatalities since early 2020 [4]. However, seasonal IAV still exists among humans. Therefore, after the COVID-19 pandemic is brought under control by SARS-CoV-2 vaccination, the number of influenza cases in association with close human-to-human contact may rise again in the absence of any protective measures [4].

**Competing interests:** The authors have declared
that no competing interests exist.

Although current influenza vaccines do not induce a sterilized immunity, they are likely to
be effective in reducing the severity of influenza [2]. The most of the current influenza vaccines
are administered parenterally. They elicit systemic IgG antibody response against IAV, but little
mucosal IgA antibody, and therefore, are insufficient to prevent infection at mucosal surfaces
[5]. Because it has been shown that mucosal IgA and IgG antibodies largely contribute to
the protection against influenza virus infection [6], a nasal route of immunization is considered
more effective for offering protection from upper respiratory infections [5]. Importantly,
a nasal immunization can induce secretory IgA that forms dimers or even multimers and
exhibits more potent and broader neutralizing activity than serum IgA monomers [7].

An intranasal vaccine containing inactivated whole virions has been demonstrated to be
immunogenic and safe since the early 2000s. However, a licensed intranasal vaccine has not
been widely utilized [8]. In the US, the nasal spray flu vaccine consisting of a live attenuated
IAV (LAIV) was approved in 2003; however, its application is limited to healthy non-pregnant
individuals aged 2–49 [9]. In the UK, a quadrivalent LAIV was introduced for children in the
2013–2014 flu season; it is now recommended for children aged 2–17 years [10]. Notably, the
effectiveness of the live IAV vaccine so far does not surpass the more widely used and safer
inactivated IAV vaccine [11, 12]. In Japan, an intranasal vaccine formula of inactivated IAV
was developed and may be available for all ages [13]. It will be interesting to know how efficacious
this intranasal vaccine can be.

To measure mucosal anti-IAV responses after intranasal IAV vaccine administration, nasal
or nasopharyngeal wash samples have been collected [13, 14]. However, collecting nasal or
nasopharyngeal washes is not easy practice and unsuitable for mass sampling. Thus, a simpler
method to evaluate the mucosal immune responses is desired for developing an intranasal IAV
vaccine [14]. The saliva sampling is much safer, easier, and less painful than nasal/nasopharyngeal
sampling; in fact, saliva tests have long been recognized as sensitive assays of viral antibodies
[15]. Furthermore, the detection rate of respiratory viruses in saliva and nasopharyngeal
aspirates are shown to be comparable [16]. Therefore, saliva collection could be an alternative
to nasal wash collection for evaluating the mucosal status after a virus infection.

As SARS-CoV-2 is frequently transmitted during meals and face-to-face conversations and
then replicated in the organs in the oral cavity [17], a saliva test is suitable for detecting SARS-
CoV-2 RNA via RT-PCR [18, 19] and comparable with nasopharyngeal swabs [20]. Thus,
saliva sampling is valuable for detecting both viral RNA and host mucosal antibody responses
in COVID-19 [21]. Moreover, detecting serum and saliva antibodies will enhance the serological
study of SARS-CoV-2 infection [22, 23] and even other upper respiratory infections. In this
context, Russel et al. have claimed that the evaluation of mucosal and serum IgA response is
important for the understanding the asymptomatic and mild states of COVID-19 [24].

In this study, we evaluated the usefulness of saliva samples for detecting mucosal immune
responses against IAV infections. With regards to the antibody class, Waldman et al. detected
IgG and IgA, but not IgM, in bronchoalveolar lavage fluids and nasal washings after mucosal
immunization of inactivated IAV vaccine [25]. In a common-cold coronavirus infection, it
was shown that IgG and IgA, but not IgM, can persist for extended periods in the serum and
nasal fluids [26]. Therefore, we here focused on only IgA and IgG antibodies. We established a
sandwich ELISA system to measure the amount of influenza virus-specific IgA and IgG in
saliva and serum. We tested the samples collected from individuals diagnosed with IAV infection
from December 2019 to February 2020 and healthy individuals before and after IAV vaccination.
We showed here that saliva samples were potentially valuable for evaluating immune
responses to influenza virus infection, especially salivary IgA (sIgA) response at an early time
point and salivary IgG (sIgG) response at a later time point. With non-invasive and safe

**Table 1. Saliva sample information of A(H1N1)pdm influenza virus-infected individuals in the 2019–2020 flu season.**

| Donor ID | Gender | Age (years) | Pre (month in 2019) | Influenza infection (month, year) | Early p.i. | Late p.i. (weeks) | Influenza history | 2019/20 vaccine |
|---|---|---|---|---|---|---|---|---|
| 2003 | F | 20 | 11 | 12. 2019 | day 8 | 5 | No | No |
| 2005 | F | 20 | 11 | 12. 2019 | day 6 | >3 | No | Yes |
| 2006 | M | 20 | 11 | 12. 2019 | day 6 | >3 | No | No |
| 2007 | M | 20 | 11 | 12. 2019 | day 9 | 4 | No | No |
| 2008 | F | 20 | 11 | 12. 2019 | day 8 | 4 | Yes | Yes |
| 2016 | M | 22 | 11 | 2. 2020 | day 8 | 4 | No | No |
| 2013 (B) | F | 21 | 11 | 1 2020 | day 8 | 5 | Yes | Yes |
| 2010 (A) | F | 21 | na | 1. 2020 | day 8 | 5 | No | No |
| 2014 (C) | F | 21 | na | asymptomatic | - | * | No | Yes |

na, not available; p.i., post-infection (1 day before the onset of high fever was set at day 0).

*, the samples were collected at the same time with A and B.

sampling, a large-scale serological screening system using saliva would be possible to study asymptomatic cases with various upper respiratory virus infections or evaluate intranasal vaccines.

## Materials & methods

### 1. Sample collection

Every year, we have been recruited student and staff volunteers from the Division of Medical Technology, Tokyo University of Technology, and obtained their serum and saliva samples with written informed consent before the influenza season (pre-stocks). During the 2019–2020 flu season from December 2019 to early February 2020, right before various measures for preventing SARS-CoV-2 transmission were widely promoted in Japan [27], there were 15 cases of influenza. They were diagnosed with IAV infection via a rapid flu test kit in local clinics using nasal swab samples. We have no influenza cases since then. Among the 15 cases, 7 complete sets of saliva samples, including the samples before infection (pre-serum), at approximately 6–10 days (early), after the documented infection (1 day before the onset of high fever was set at day 0), and at 3–5 weeks after infection, were available (Table 1, down to no.2013). Unfortunately, the pre-serum samples were unavailable in 5 of 7. The serum and saliva samples of two additional individuals were also analyzed, who had close contact with individual no. 2013 (B in Table 1) just before the influenza onset; one infected no. 2010 and the other asymptomatic no.2014 (A and C in Table 1, respectively).

Also, we used the saliva samples of 11 individuals who received an influenza vaccine during the 2018–2019 flu season completely different from influenza-infected individuals; the samples were stocked before (pre) and approximately 1 month after (post) vaccination (Table 2).

Saliva samples were collected using SalivaBio (Salimetris, CA, USA) oral swabs and centrifuged at $1,710 \times g$ for 15 min, and the supernatants were transferred to new tubes. All the samples were kept in a −80˚C freezer until use. The study followed the Helsinki declaration and was approved by the ethical committee of the Tokyo University of Technology (No. E18HS-023).

### 2. Reagents and viruses

An inactivated influenza virion of A(H1N1)pdm09 vaccine strain [A/Singapore/GP1908/2015 (H1N1)] was kindly provided by Dr. Takeshi Tanimoto (BIKEN Co. Ltd., Kagawa, Japan). The

**Table 2. Saliva Sample information of vaccinated individuals in the 2018–2019 flu season.**

| Donor ID | Gender | Age | Pre (month. year) | Post-vaccination (weeks) | Influenza history | Vaccination history |
|---|---|---|---|---|---|---|
| 19V1 | M | 20 | 10. 2018 | 4 | Yes | Yes |
| 19V2 | F | 21 | 10. 2018 | 4 | Yes | Yes |
| 19V3 | F | 21 | 11. 2018 | 4 | Yes | No |
| 19V4 | F | 21 | 11. 2018 | 4 | No | No |
| 19V5 | F | 21 | 11. 2018 | 4 | No | Yes |
| 19V6 | F | 22 | 11. 2018 | 6 | No | No |
| 19V7 | F | 22 | 11. 2018 | 6 | No | No |
| 19V8 | F | 22 | 11. 2018 | 8 | No | Yes |
| 19V9 | M | 67 | 11. 2018 | 5 | No | Yes |
| 19V10 | M | 66 | 11. 2018 | >3 | No | Yes |
| 19V11 | M | 21 | 11. 2018 | 4 | Yes* | Yes |

*Infected with influenza B virus in February 2018.

live H1N1 (A/Singapore/GP1908/2015) and H3N2 (A/Hong Kong/4801/2014) viruses were obtained from the Influenza Virus Research Center, National Institute of Infectious Diseases (NIID). These viruses were expanded in MDCK (ATCC) cells, and their respective titers were determined using the hemagglutination activity (HA) assay with type O human red blood cells (RBCs). The H1HA, HA protein of H1N1 (A/Singapore/GP1908/2015) virus was produced by a baculovirus expression system (Thermo Fisher Scientific, MA, USA).

## 3. Generation of standard influenza-specific antibodies

FI6 is a well-known, broadly IAV-reactive human monoclonal antibody [28]. The expression plasmids of the $V_H$ and $L_H$ gene of the FI6-IgG antibody were constructed using a recombinant technology with a human antibody expression cassette [29]. The constant region of the IgA heavy chain gene cloned from the peripheral blood mononuclear cells of a healthy donor was used to generate an FI6-IgA-expression plasmid by replacing the genes encoding the constant region of IgG with that of IgA using Gibson assembly (New England Biolab., MA, USA). The recombinant FI6-IgG and FI6-IgA antibodies were produced using the Expi293$^{TM}$ Expression System Kit (Thermo Fisher Scientific). FI6-IgG and FI6-IgA were purified with Protein G Sepharose 4B (Thermo Fisher Scientific) and Peptide M/agarose (InvivoGen Inc., San Diego, CA, USA), respectively.

## 4. Hemagglutination inhibition (HI) test

The HI test was performed using live H1N1 (A/Singapore/GP1908/2015) and H3N2 (A/Hong Kong/4801/2014) viruses according to the standard protocol provided by WHO (https://www.who.int/influenza/gisrs_laboratory/cnic_serological_diagnosis_hai_a_h7n9_20131220.pdf) with slight modifications. In summary, each serum was first pretreated with RDE to remove non-specific hemagglutinin inhibitors and pre-adsorbed with type O human RBCs. Then, the sera were serially diluted 1:2 starting from a 1:20 diluted solution with 25 μL/well in a 96-well conical-bottom plate (Watson Bio Lab, Tokyo, Japan). Next, the diluted sera were incubated with four HA influenza virus titers at 25 μL/well for 1 h at room temperature (RT: 25°C) and then with 0.75% of human RBCs at 50 μL/well for 1 h at 4°C. The minimum HI titer was 20.

## 5. ELISA

The amount of total IgA in saliva was measured. First, a Nunc 96-well microtiter plate (Thermo Fisher Scientific) was coated with a mouse monoclonal anti-human-IgA antibody (22C) [30] at 1 µg/mL and kept at 4°C overnight. The next day, the plate was washed, blocked with PBS/1% BSA at RT for 1.5 h before the samples were added. The samples were first serially diluted 5-fold with PBS/1% BSA/0.1% Tween-20 starting from a 1:100 dilution solution. As a standard, purified myeloma $IgA_1$ protein (Sigma-Aldrich Inc., Tokyo, Japan) was used in 1:2 serial dilutions starting from 20 ng/mL. Then, the plate was placed at 4°C overnight, washed again, and incubated with a biotinylated anti-human IgA antibody (Southern Bio-Tech, Birmingham, UK). After a 1-h incubation at RT, the plate was washed and reacted with HRP–streptavidin (1:2000 dilution with PBST, BioLegend) for 30 min. Afterward, the TMB substrate (Sigma-Aldrich) was added after washing. Last, the reaction was stopped and measured at $OD_{450}$ using a microplate reader $iMARK^{TM}$ (BioRad, Hercules, CA, USA).

The amount of influenza-specific IgA and IgG in each sample was measured using the aforementioned ELISA protocol, except the plate was coated with inactivated A(H1N1)pdm09 virions (1:2000 dilution) or H1HA protein (2 µg/mL) in PBS. Purified recombinant FI6-IgA or FI6-IgG antibody was used as a standard to measure virion-specific IgA or IgG. In some experiments, a purified human IgG against H1HA was used to measure H1HA-specific IgG. Saliva samples were diluted serially 1:3 starting from a 1:20 dilution solution and serum samples were diluted serially 1:5 starting from a 1:100 dilution solution. The secondary antibody used was a biotinylated anti-human IgA or IgG antibody (Southern Bio-Tech).

## 6. Statistical analysis

The amount of A(H1N1)-specific IgA and IgG antibodies was calculated on the basis of a standard curve of FI6-IgA and FI6-IgG, respectively, using Microplate Manager 6 (BioRad). Statistical analysis was performed using GraphPad Prism version 9. The differences between groups were analyzed using the nonparametric Mann–Whitney test or Wilcoxon matched-pairs test. A P-value <0.05 was considered statistically significant.

## Results

### 1. HI titer and anti-H1HA IgG antibodies in the serum

From December 2019 to early February 2020, the A(H1N1)pdm09 influenza virus caused a major epidemic in Japan; however, A(H3N2) influenza infections were rare [23]. Therefore, we first conducted the HI assay to analyze the serum of seven individuals diagnosed with IAV infection (Table 1) to determine the viral strain. Although the pre-serum was unavailable in five of the seven individuals, we detected >80 HI titers against A(H1N1)pdm09 in all the late (3–5 weeks after infection) samples except no. 2008 (Fig 1A., * no sample available). Notably, donor nos. 2003, 2006, 2007, and 2016 were never vaccinated. In contrast, donor nos. 2005, 2008, and 2013 were vaccinated every year, including the 2019–2020 flu season. Concerning the HI titer against the A(H3N2) influenza virus, it was found <40 in all the serum samples.

We also measured the amount of H1HA [A/Narita/1/2009(H1N1)]-specific serum antibodies via ELISA. The results were roughly consistent with those of HI titers; also, a slight increase in the level of specific IgG was detected in donor no. 2008 (Fig 1B). Because the "pre" serum was available only in no. 2016 and no. 2013, it is not clear whether the relatively high basal level of HA-specific serum IgG in no. 2016 is within the variability of ELISA measurement or due to some other reasons. Collectively, the data suggest that all seven individuals were infected with the A(H1N1)pdm09 influenza virus.

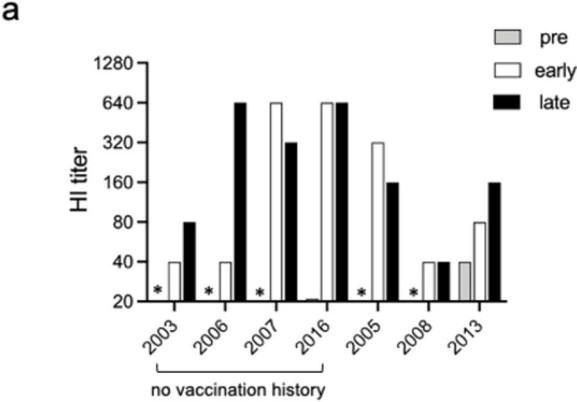

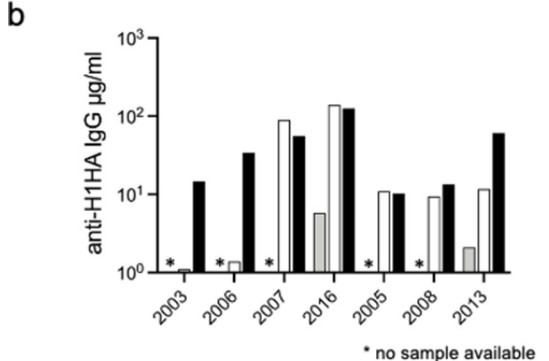

**Fig 1. The titer of Hemagglutination inhibition (HI) and serum IgG in influenza cases in the 2019–2020 flu season.** Serum samples of seven donors diagnosed with influenza A virus infection were analyzed. (a) Standard HI assay. The y-axis shows the HI titers. (b) ELISA. The amount of IgG was calculated on the basis of the standard curve of a purified H1HA-specific human IgG antibody. Grey columns and symbols, before infection; blank columns, 8–10 days after infection (early); black columns and symbols, >1 month after infection (late).

## 2. The influenza-specific IgA increases earlier than the influenza-specific IgG in saliva

Because the level of secretory IgA in the saliva is high and known to vary over a day [28], we simultaneously measured the total amount of sIgA and calculated the portion of influenza-specific IgA. The amount of total sIgA varied from 11.8 to 65.5 µg/mL, and the ratio of the early and late samples to the pre sample was 0.261–4.274. The portion of influenza-specific sIgA increased at the early sample in five (nos. 2007, 2016, 2005, 2008 and 2013) of the seven individuals and decreased significantly later, based on the pre-infection titers (fold increase in Fig 2A, right panel). In contrast, the fold increase of specific sIgG tended to be higher later (Fig 2B) than early. Notably, in the individuals (nos. 2003 and 2006) who showed little increase in sIgA even in the late time point, the level of sIgG increased especially in the late time point (Fig 2A and 2B, right panels). To make the time course difference clearly visible, the fold sIgA and sIgG increases were summarized according to the early and late time points in Fig 2C. Although the later decline of sIgA was not significant (p = 0.078), the level of sIgG was significantly increased later (p = 0.031).

## 3. No increase of the influenza-specific salivary IgA in vaccinated individuals

Parenteral vaccination does not significantly induce mucosal IgA responses [5]. We verified this observation by analyzing the A(H1N1)pdm09-specific sIgA in 11 vaccinated individuals

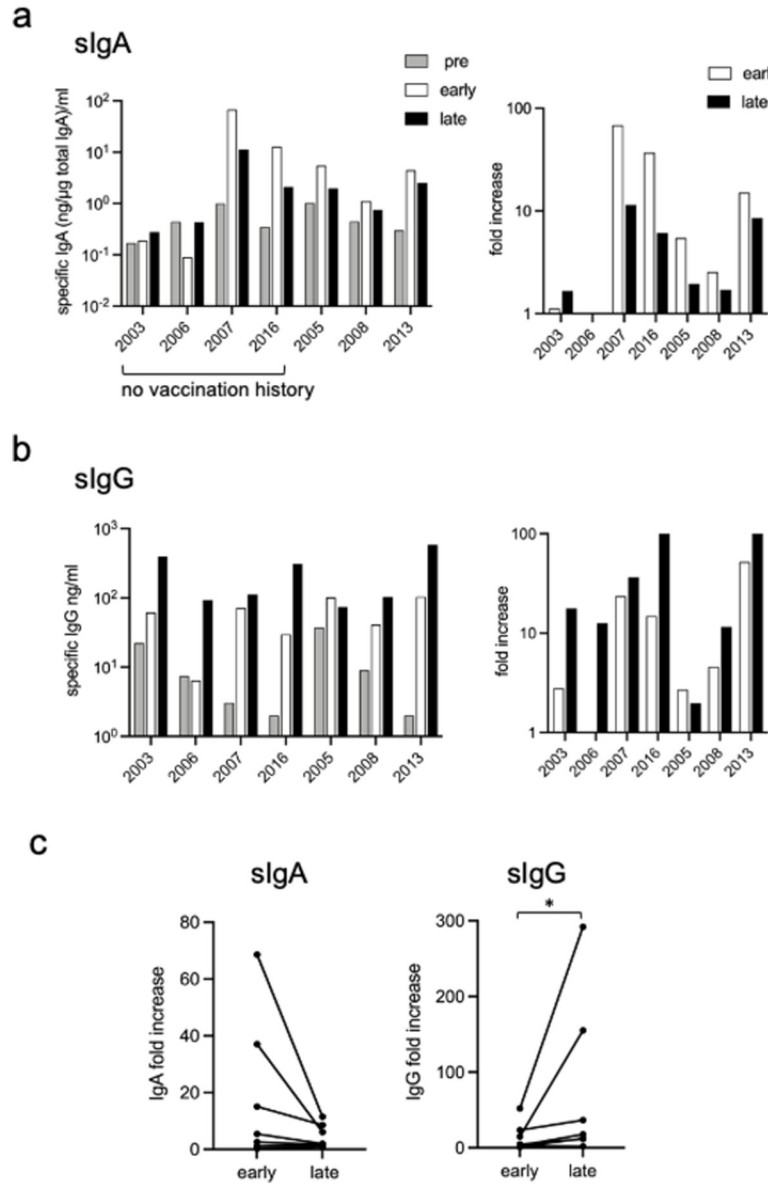

**Fig 2. The level of saliva antibodies in A(H1N1)pdm influenza virus-infected individuals.** Total sIgA, A(H1N1) pdm virion-specific sIgA, and sIgG were determined using ELISA. (a) The calculated amount of specific sIgA (ng) per 1 μg of total sIgA is depicted on the y-axis. Right panel: the fold increases from pre-antibody titers. (b) The y-axis indicates the amount of specific sIgG (ng). Right panel: the fold increases from pre-antibody titers. Grey column, before infection (pre); blank column, 8–10 days after infection (early); black column, 3–5 weeks after infection (late). (c) The fold sIgA and sIgG increases in the right panels were summerized according to the early and late time points. *P < 0.05.

(Table 2) before (pre) and after (post) inoculation with the 2018–2019 flu vaccine. A very low level of A(H1N1)pdm09-specific sIgA was detected (Fig 3A, left panel); the sIgA levels remained largely unchanged after vaccination except in V11. The mean titers in all the pre and post samples were similar, at 0.485 and 0.504 ng/μg total sIgA per milliliter, respectively. In some donors, the titer was even decreased after vaccination. Thus, basically, the sIgA was not increased by vaccination. Of note, the basal level of specific sIgA tended to be higher in three individuals (Group A: V01–V03) who had a previous history of influenza >3 years ago

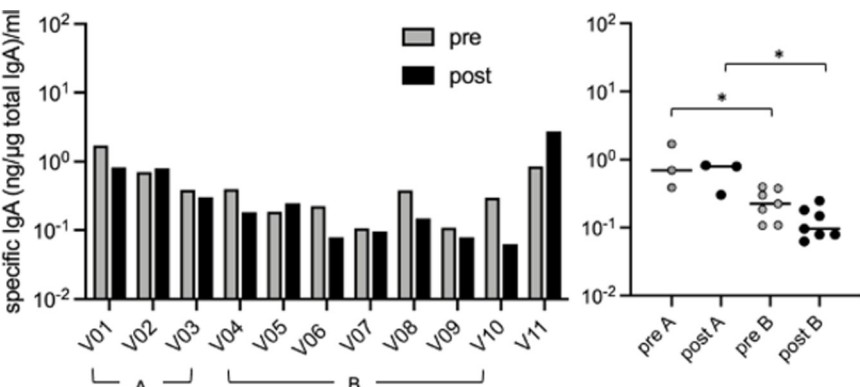

**Fig 3. The level of saliva antibodies in vaccinated individuals.** The saliva samples were collected before (pre) and after (post) the 11 individuals received the influenza vaccine. Total sIgA and A(H1N1)pdm-specific sIgA were determined using ELISA, and influenza virus-specific sIgA (ng) per total sIgA (μg) was calculated as described in Fig 2. Right panel: the difference between Group A (n = 3) with a previous history of influenza and Group B (n = 7) without the history of influenza was depicted. Donor V11, who had a recent influenza B virus infection, was excluded from the comparison. Grey column, before infection (pre); black column, 3–5 weeks after infection (late). *P < 0.05.

compared with that in the seven individuals (Group B: V04–V10) who had no influenza history. Because donor V11, who was infected with an influenza B virus 9 months before vaccination, displayed a considerable increase in the level of sIgA after vaccination, we excluded the person from the comparison between Groups A and B. Nevertheless, the sIgA levels were significantly higher in Group A than those in Group B in both the pre- and post-vaccination samples (p < 0.033 and p < 0.022, respectively) (Fig 3, right panel).

## 4. Symptomatic and asymptomatic infection in contact cases

Finally, we present an example of infection caused by the close contact among three female students. One student (A) developed influenza in the first day of January, and the other (B) developed influenza 1 day later. Both were diagnosed with influenza A. The third individual (C) never developed any symptoms. The level of serum H1HA-specific IgG was increased in A and B (Fig 4A), whereas it was already high in C before the incidence and only increased slightly later.

The levels of sIgA and sIgG were measured in these three individuals (Fig 4B). The saliva sample of C was collected with that of A and B only at the later time point. Although A's pre-sample was unavailable, the level of A(H1N1)pdm09-specific sIgA was remarkably higher in A than in B at the early time point (Fig 4B, left).

Interestingly, C had a substantial level of sIgA after the contact incidence. A similarly high level of sIgG was detectable in all individuals at the late time point (Fig 3B, right). A had never been vaccinated, whereas B and C were vaccinated every year. Of note, C received a flu vaccine just 1 week before the incident, and her pre-serum sample was obtained 1 month before the vaccination. Therefore, the slight increase in the level of serum IgG in C at the late time point could be due to the vaccination. Conversely, a substantial level of sIgA may reflect a transient exposure to an influenza virus.

## Discussion

We showed the usefulness of saliva as a biomarker of the mucosal immune status during an upper respiratory infection by analyzing the sIgA and sIgG responses to influenza virus infection. We collected samples from individuals who were clinically diagnosed with influenza type

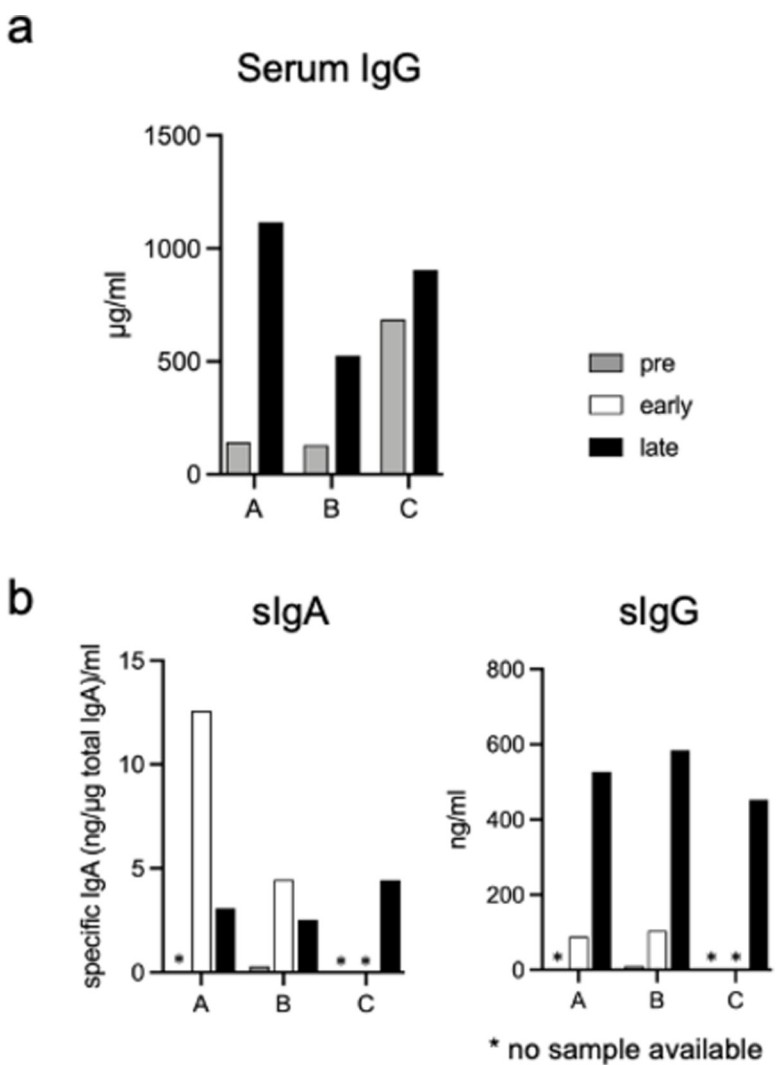

**Fig 4. Serum and saliva antibodies in close contact cases.** (a) The level of serum anti-H1HA IgG before (pre) and 1 month after infection incidence (late) in three donors. (b) The level of sIgA (left) and sIgG (right) of these individuals (ng/mL). Grey columns: before infection (pre); blank columns, 8–10 days after infection (early); black columns, >1 month after infection (late). *No sample was available.

A and confirmed that they were infected with the A(H1N1)pdm09 influenza virus. Since those patients are treated with an anti-influenza drug within 48 h after the onset of symptoms, saliva samples may not fully reflect the natural course of influenza virus infection. Nevertheless, we could detect the early increase (at 6–10 days post-infection) of sIgA and later increase (at 3–5 weeks) of sIgG. The mucosal immune system consists mainly of nasopharynx-associated lymphoid tissues (NALT) and gut-associated lymphoid tissues (GALT) and is partly compartmentalized depending on the actual route of induction [24]. It is noted that parotid secretory IgA in saliva could be linked to immune induction in tonsils/adenoid (human NALT) and cervical lymph nodes than that in GALT [31]. In contrast, most IgG in saliva has been considered to be derived from the blood circulation by passive leakage [31]. In fact, the plasma and salivary IgG profiles are shown to be highly similar [32]. Therefore, a later increase of sIgG likely reflects a systemic IgG response. Thus, despite the small sample size, this is the first important study

indicating that saliva sampling is quite valuable for the analysis of mucosal anti-influenza response in parallel with a systemic response.

The level of secretory IgA is one of the key parameters in evaluating mucosal vaccination. The representative mucosal antibodies after intranasal vaccination can be obtained by nasal washing with 100 mL of PBS [13, 14], a non-invasive but unpleasant and even painful procedure. Thus, nasal wash sampling is unsuitable as a routine procedure for ordinary people. In contrast, saliva collection is simple, easy, and painless. In addition, analyzing salivary IgA represents an easy method of measuring immunogenicity after administering intranasal vaccination with LAIV to children [33]. Furthermore, oral fluid sampling has been recently validated for assessing responses to LAIV vaccine [34]. Our data indicate that saliva can serve as a simple indicator of mucosal immune responses to upper respiratory virus infections. Moreover, saliva tests may help evaluate and develop mucosal vaccines [8, 14, 35].

Parenteral vaccination is considered to induce little IgA in the mucosa, including saliva [8]. Consistent with this notion, the current parenteral flu vaccine did not induce A(H1N1)pdm09 virus-specific IgA in the saliva in this study. One exception was donor V11 (Fig 3), who was infected with influenza B virus in February 2019, 9 months before flu vaccination. As antibodies against the A(H1N1)pdm09 influenza virus persist for at least 15 months [36], the residual mucosal immune response against some common antigens of influenza viruses may be reactivated through vaccination. Interestingly, the basal level of A(H1N1)pdm09 virus-specific IgA was relatively higher in the three donors with a previous history of influenza than that in the seven donors without a history of influenza. Thus, the result suggests that while the subcutaneous vaccination failed to induce mucosal antibody responses, the primed individuals by a previous IAV infection can maintain the mucosal antibody to some extent. However, the levels of influenza-specific IgA are too low for us to conclude. Further study with larger sample size is necessary.

The protective role of mucosal IgA against upper respiratory infections has long been contested by challenge studies using animal models and humans [37]. In fact, the A(H1N1) challenge study by Gould et al. of individuals with only low HI titers showed that H1N1 specific IgG levels did not correlate with protection; meanwhile, specific IgA levels shortened the duration of virus shedding, indicating that IgA contributed to protection against influenza [38]. By analyzing mucosal antibody responses in saliva of three individuals who were simultaneously exposed to the influenza virus one (case C in Fig 4) of the three persons received vaccination a week ago and remained asymptomatic. Her serum IgG level appeared to be already high compared with that of others when they were exposed to an influenza virus. Interestingly, both sIgA and sIgG responses were detected in this individual at the late time point after the exposure. A high level of sIgA might indicate that the mucosal antibodies are induced by brief exposure to an influenza virus and the serum IgG induced by vaccination reflects the level of sIgG as demonstrated by Hettegger et al. that plasma and salivary anti-hepatitis B IgG profiles are highly similar for each individual infected with hepatitis B virus [32]. However, it remains unclear here which mucosal IgA and IgG antibodies play a major protective role. The analysis of more contact infection cases may shed light on the protective effect of mucosal antibody responses at an early phase of upper respiratory infections.

The induction and longevity of IgA-type antibody-secreting and memory B cells in the upper respiratory virus infection remains poorly understood. Interestingly, in the study of respiratory syncytial virus (RSV) challenge, Habiti et al. detected influenza-specific IgA memory B cells in the peripheral blood circulation [39]. Because humans are not naïve to IAV infection, it will be challenging to study the generation/evolution of upper respiratory memory IgA B-cell responses in influenza. The recently emerged SARS-CoV-2 virus infection may provide a good opportunity to clarify this issue. In this context, a wave of IgA plasmacells in the blood

is considered to occur prior to the production of secretory IgA antibodies, with a peak at around 6–10 days after mucosal infection/immunization [24]. Recently, Sterlin et al. reported that early SARS-CoV-2-specific humoral responses were dominated by IgA antibodies and IgA plasmablasts with mucosal homing potential was detected early [40]. It is an interesting question to address how and where such mucosal homing IgA plasmablasts and memory cells are induced and maintained in humans.

## Conclusions and perspectives

Serum antibody responses are mostly measured in serological studies in epidemiology or evaluations of protection by vaccines. However, in respiratory infections, which occur primarily in the upper respiratory mucosa, both mucosal and serum antibody responses should be measured in parallel. Many questions remain unanswered in the study of influenza. For example, how frequently do asymptomatic influenza infections occur? How are influenza viruses maintained in humans during summertime? Most importantly, what is the true correlate of protection in vaccination? Large-scale serological studies will be critical for answering these questions. In that sense, our results strongly support the notion that measuring salivary IgA and IgG in addition to serum IgG is a promising way of mass screening the intranasal vaccination status or identifying asymptomatic infections of influenza and other respiratory viruses including SARS-CoV-2 [24]. The usefulness of saliva has already been noted in SARS-CoV-2 infection and other virus infections. How saliva testing contributes to understanding the human respiratory immune responses needs to be studied further.

## Supporting information

**S1 Data.**
(XLSX)

## Acknowledgments

We thank the volunteer staff and students in the Department of Medical Technology, School of Human Sciences, Tokyo University of Technology, for providing saliva and serum samples and narration of infection/vaccination history in detail. The sampling and antibody measuring was done partly by the active cooperation of those students who learn technologies for clinical laboratories. We also thank Dr. Takeshi Tanimoto (BIKEN, Osaka University) for providing inactivated virions of all influenza vaccine strains. In addition, we appreciate helpful comments and suggestions by Drs. Shigeyuki Itamura and Hideki Asanuma (Influenza Virus Research center, NIID). The authors would like to thank Enago (www.enago.jo) for the English language review.

## Author Contributions

**Conceptualization:** Yasuko Tsunetsugu-Yokota.

**Data curation:** Yasuko Tsunetsugu-Yokota.

**Formal analysis:** Yasuko Tsunetsugu-Yokota.

**Investigation:** Yasuko Tsunetsugu-Yokota, Sayaka Ito, Taishi Onodera.

**Methodology:** Yasuko Tsunetsugu-Yokota, Sayaka Ito, Yu Adachi, Taishi Onodera, Tsutomu Kageyama.

**Project administration:** Yasuko Tsunetsugu-Yokota.

**Resources:** Yu Adachi, Taishi Onodera, Tsutomu Kageyama, Yoshimasa Takahashi.

**Supervision:** Yoshimasa Takahashi.

**Validation:** Tsutomu Kageyama, Yoshimasa Takahashi.

**Visualization:** Yasuko Tsunetsugu-Yokota.

**Writing – original draft:** Yasuko Tsunetsugu-Yokota.

**Writing – review & editing:** Yasuko Tsunetsugu-Yokota, Yu Adachi, Tsutomu Kageyama, Yoshimasa Takahashi.

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
