## [Decision Letter · Decision Letter 0]

19 Oct 2021

PONE-D-21-28281Saliva as a useful tool for evaluating upper mucosal antibody response to influenzaPLOS ONE

Dear Dr. Tsunetsugu-Yokota,

Thank you for submitting your manuscript to PLOS ONE. After careful consideration, we feel that it has merit but does not fully meet PLOS ONE’s publication criteria as it currently stands. Therefore, we invite you to submit a revised version of the manuscript that addresses the points raised during the review process.

We look forward to receiving your revised manuscript.

Kind regards,

Paulo Lee Ho, Ph.D.

Academic Editor

PLOS ONE

Journal Requirements:

Reviewers' comments:

Reviewer's Responses to Questions

**Comments to the Author**

1. Is the manuscript technically sound, and do the data support the conclusions?

Reviewer #1: Yes

Reviewer #2: Yes

2. Has the statistical analysis been performed appropriately and rigorously? 

Reviewer #1: Yes

Reviewer #2: No

3. Have the authors made all data underlying the findings in their manuscript fully available?

Reviewer #1: Yes

Reviewer #2: Yes

4. Is the manuscript presented in an intelligible fashion and written in standard English?

Reviewer #1: Yes

Reviewer #2: Yes

5. Review Comments to the Author

Reviewer #1: The theme has scientific relevance and has a well-structured methodology, unfortunately it has a small sample size, however, this does not minimize the merit of the study.

The introduction presents the topic clearly and objectively. The methodology is adequate. The presentation of the results is clear, however it presents poor quality figures. Regarding the discussion, this could be more clear and fluid in reading, the paragraphs are without continuity of reasoning. The first paragraph could be more aimed at presenting the novelty in the study's findings, with fewer descriptions and without going back to introductory or methodological issues. The literature review seems adequate enough to support the discussion.

Overall, a structural review of the discussion could enrich the presentation of this study and consequently improve its quality.

Reviewer #2: According to the authors, the objective of this study, in a general way, was to evaluate the potential use of saliva to measure the amount of influenza virus-specific IgA and IgG antibodies by using a quantitative ELISA "in house". Interesting data were presented and they can support the putative use of saliva samples to monitor the influenza virus infection or vaccination by using the quantitative ELISA test purposed by the authors. However, it is necessary to clarify that the main limitation of the study was the low number of samples evaluated. In addition, there are another couple of factors that should be concerned, too.

In the Abstract section:

1) Please describes the salivary IgA and IgG as secretory, which allows us to differ from serum IgA and IgG.

2) Please state in which sample the results concerning early IgA and latter IgG were found. It is not clear if these results were observed only in saliva or only in serum or in both fluids.

3) Since IgM is an antibody that can be secreted by the mucosa, including in the upper airways, why IgM levels were not evaluated? If was measured, it could be interesting to report these data.

4) Please state what was the volunteer groups enrolled in this study since was not clear the reason to cite vaccination for the Influenza virus or why the authors reported a result concerning one asymptomatic individual.

In the Introduction section:

5) The authors declare that nasal and saliva could be useful to evaluate the antibodies levels in the upper airways. However, it is noteworthy to mention that, whereas secretory IgA (SIgA), SIgM, and IgG can be detected in saliva samples, IgG is not easily detected in nasal fluids. Therefore, it is recommended to highlight this fact in the "Introduction" section in order to reinforce the use of saliva to monitor these antibodies responses in the study context.

In the Material and Methods section:

6) Please state that the study followed the Helsinki declaration and also it is necessary to present the study approved number from Ethics Committee.

7) How the serum was obtained?

8) Please clarify whether the samples used in the group vaccinated were obtained in a totally different group from the influenza-infected group

9) Since some analyses were performed with samples obtained on three different occasions, Friedman's test with Dunn's post hoc test should be used.

In the Results section

10) In order to be clearer, I suggest reorganizing the following sentences on page 12, lines 185-188, as described below.

"Notably, donor nos. 2003, 2006, 2007, and 2016 were never vaccinated. In contrast, donor nos. 2005, 2008, and 2013 were vaccinated every year, including the 2019–2020 flu season. Concerning the HI titer against the A(H3N2) influenza virus, it was found <40 in all the serum samples."

11) The authors did not report that the specific IgG levels for HIHA from the volunteer 2006 were increased in time point "pre" (Fig 1b). This is an interesting result since the HI titer of this volunteer was under the detection rate at the same time point (Fig. 1a).

12) Please remove the last sentence on page 12, lines 195-196, and the first paragraph of page 13, lines 197-200, due to the fact that these pieces of information were already presented in the "introduction" section.

13) Although the suggestion presented on page 13, lines 209-211, are relevant, I believe that these pieces of information could be useful in to "Discussion" section. So, please remove it from the "Results" section to the "Discussion" section.

14) In Fig. 3a it is possible to observe that the values of A(H1N1)pdm09-specific salivary IgA from the volunteer V11 were increased post-vaccination. Corroborating this observation the authors cited, on page 14, lines 217-218, that "...the IgA levels remained largely unchanged after vaccination except in V11." So, I would like to know whether the values obtained pré and post-vaccination were significantly different?

15) Are there statistical differences in the results presented in Figs. 4a and 4b?

16) Again, on page 14, lines 227-230, as well as on page 15, lines 242-247, the authors presented a putative conclusion or even suggestions for the results found. I recommend removing these sentences since this section must only present the results obtained in the study.

In the Discussion section:

17) The authors discuss, in a very well way, the potential use of saliva for monitoring not only the influenza virus infection but also its annual vaccination. However, some points could be improved:

- Although saliva could be considered a "corollary" fluid to evaluate the mucosal immunity of upper airways, there are some specific differences in oral and nasal mucosa immunity that must be discussed. Therefore, I suggest reorganizing some pieces of information from the "Introduction" section to the "Discussion" section.

- In addition, the "time" of mucosal immunity response should be more discussed since this fact could impact the evaluations presented in this study. For instance, as recently mentioned by Dos Santos et al.*, "...the literature demonstrates that SIgA is an important tool for early detection of infections, normally, 1 day after the infection, whereas serum IgA and IgM can be detected after 3–5 days after the infection (30)".

* Dos Santos JMB, Soares CP, Monteiro FR, Mello R, do Amaral JB, Aguiar AS, Soledade MP, Sucupira C, De Paulis M, Andrade JB, Almeida FJ, Sáfadi MAP, Mau LB, Brasil JM, Ramalho T, Loures FV, Vieira RP, Durigon EL, de Oliveira DBL, Bachi ALL. In Nasal Mucosal Secretions, Distinct IFN and IgA Responses Are Found in Severe and Mild SARS-CoV-2 Infection. Front Immunol. 2021 Feb 25;12:595343. doi: 10.3389/fimmu.2021.595343. eCollection 2021.

6. PLOS authors have the option to publish the peer review history of their article (what does this mean?). If published, this will include your full peer review and any attached files.

Reviewer #1: No

Reviewer #2: **Yes: **ANDRE LUIS LACERDA BACHI

---

## [Author Response · Author response to Decision Letter 0]

29 Nov 2021

Responses to reviewers

Reviewer #1: The theme has scientific relevance and has a well-structured methodology, unfortunately it has a small sample size, however, this does not minimize the merit of the study.

The introduction presents the topic clearly and objectively. The methodology is adequate. The presentation of the results is clear, however it presents poor quality figures. Regarding the discussion, this could be more clear and fluid in reading, the paragraphs are without continuity of reasoning. The first paragraph could be more aimed at presenting the novelty in the study's findings, with fewer descriptions and without going back to introductory or methodological issues. The literature review seems adequate enough to support the discussion.

Overall, a structural review of the discussion could enrich the presentation of this study and consequently improve its quality.

- Thank you for your kind and constructive comments. We have uploaded the figures with better resolution and added a new Fig.2c. 

- We completely agree with you that the first paragraph has too much redundancy. We have extensively reorganized the discussion part. We hope that you find now the appropriate presentation of our study in the discussion of the revised manuscript.

 

Reviewer #2: According to the authors, the objective of this study, in a general way, was to evaluate the potential use of saliva to measure the amount of influenza virus-specific IgA and IgG antibodies by using a quantitative ELISA "in house". Interesting data were presented and they can support the putative use of saliva samples to monitor the influenza virus infection or vaccination by using the quantitative ELISA test purposed by the authors. However, it is necessary to clarify that the main limitation of the study was the low number of samples evaluated. In addition, there are another couple of factors that should be concerned, too.

- Thank you very much for your helpful comments. We admit the limitation of this study on the low number of samples. However, these samples are valuable especially because we were able to use “pre” saliva samples before A(H1N1)pdm influenza infection. They are essential to evaluate the fold antibody increase at early and late time points. As reviewer #1 commented, we believe that a small sample size will not minimize the merit of the study, and our results are worth publishing. We have claimed this point in the last part of first paragraph in the discussion.

In the Abstract section:

1) Please describes the salivary IgA and IgG as secretory, which allows us to differ from serum IgA and IgG.

- Thank you for your suggestion. To discriminate serum and saliva antibodies, we use sIgA and sIgG for salivary antibodies in the abstract and a whole text. 

2) Please state in which sample the results concerning early IgA and latter IgG were found. It is not clear if these results were observed only in saliva or only in serum or in both fluids.

- Thank you for your comment. Our major finding using saliva samples is illustrated in Fig. 2, where the fold increase of sIgA (Fig. 2a) and sIgG (Fig. 2b) were depicted at the right panels. As you see, early increase (blank column) was observed in 5 of 7 individuals, whereas in all individuals except no. 2015, the fold increases of sIgG were higher at later (filled column) than early time point. To make the time course difference clearly visible, the fold sIgA and sIgG increases were summarized according to the early and late time points in Fig. 2c (new).

3) Since IgM is an antibody that can be secreted by the mucosa, including in the upper airways, why IgM levels were not evaluated? If was measured, it could be interesting to report these data.

- Thank you for your valuable comment. We understand your interest in mucosal IgM class antibodies. However, Waldman et al. detected IgG and IgA, but not IgM, in bronchoalveolar lavage fluids and nasal washings after mucosal immunization of inactivated IAV vaccine (Ref. [25]). Also, in a common-cold coronavirus infection, it was shown that IgG and IgA, but not IgM, can persist for extended periods in the serum and nasal fluids (Ref. [26]). Therefore, we focused on IgA and IgG antibodies in saliva and did not measure the IgM antibody. We have added this statement in the last paragraph of the introduction.

4) Please state what was the volunteer groups enrolled in this study since was not clear the reason to cite vaccination for the Influenza virus or why the authors reported a result concerning one asymptomatic individual.

- Thank you for your comment. 

- For sampling, the volunteers have been recruited in the department of medical technology where the first author belongs. Because these students have to learn how to take blood from each other and use their serum for several clinical tests as training, their serum samples were stocked every year. When they practice in the hospital at some period, they are requested to have an influenza vaccine. Taking this advantage, we planned to collect saliva and serum samples on vaccination and on infection, which was approved by the ethical committee. Since 2017, the students who got influenza have been called for cooperation. The year 2019/2020 season was a rare occasion that many students whose saliva samples were stocked were infected with H1N1 influenza. Furthermore, we had an opportunity to know the situation of close contact infection in detail among 3 students. We believe that this study could only be done in this special situation and has led us to important and suggestive findings.

- I added the following sentence in the Acknowledgments; The sampling and antibody measuring was done partly by the active cooperation of those students who learn technologies for clinical laboratories.

In the Introduction section:

5) The authors declare that nasal and saliva could be useful to evaluate the antibodies levels in the upper airways. However, it is noteworthy to mention that, whereas secretory IgA (SIgA), SIgM, and IgG can be detected in saliva samples, IgG is not easily detected in nasal fluids. Therefore, it is recommended to highlight this fact in the "Introduction" section in order to reinforce the use of saliva to monitor these antibodies responses in the study context.

- Thank you for your valuable comment. I did not know that IgG is not easily detected in nasal fluids. As I described in the response of 3), in a common-cold coronavirus infection, IgG and IgA responses persisted for extended periods in the serum and nasal fluids (Ref. no. 26]). Also, please note that in the nasal vaccine studies, both HA-specific IgG and IgA were detectable in the nasal wash, though the level of IgG is lower than that of IgA (Ref. no. 12, 13). Because the extensive concentration of a large amount of nasal wash (~100 ml) was required for measuring these antibody titers, the saliva sampling has great advantage as I described in the introduction and also in discussion. 

In the Material and Methods section:

6) Please state that the study followed the Helsinki declaration and also it is necessary to present the study approved number from Ethics Committee.

- The approved number from the ethical committee is given on page 9 of the revised manuscript.

7) How the serum was obtained?

- As we commented in 4), blood sampling is a routine task as a medical technologist. In this study, 2~3 ml blood was taken either by a medical doctor or students under the supervision of a medical doctor in the university. 

8) Please clarify whether the samples used in the group vaccinated were obtained in a totally different group from the influenza-infected group

- Yes, the saliva samples of the group vaccinated are collected and stocked during the 2018/2019 flu season (Table 2). It is completely different individuals from the influenza-infected in 2019/2020 flu season (Talbe1).

- We added a phrase on page 8.

9) Since some analyses were performed with samples obtained on three different occasions, Friedman's test with Dunn's post hoc test should be used.

- Thank you for your comment. You may have meant “three different occasions” as pre, early, and late time points in Fig. 2. However, individuals have variable basal antibody titer before IAV infection. So, basically, the pre titer needs to be used to assess the increased level of antibody in virus infection, which is a “fold increase” as depicted in the right panel. Because Friedman’s test is a nonparametric test that compares three or more paired groups, we did not apply it here.

In the Results section

10) In order to be clearer, I suggest reorganizing the following sentences on page 12, lines 185-188, as described below.

"Notably, donor nos. 2003, 2006, 2007, and 2016 were never vaccinated. In contrast, donor nos. 2005, 2008, and 2013 were vaccinated every year, including the 2019–2020 flu season. Concerning the HI titer against the A(H3N2) influenza virus, it was found <40 in all the serum samples."

- Thank you for your helpful suggestion. We amended this part according to your suggestion.

11) The authors did not report that the specific IgG levels for HIHA from the volunteer 2006 were increased in time point "pre" (Fig 1b). This is an interesting result since the HI titer of this volunteer was under the detection rate at the same time point (Fig. 1a).

- The reviewer may point out the relatively high level of HA-specific IgG before IAV infection in no. 2016 (not 2006) serum with no HI activity. We added the following comment on page 13; “Because the “pre” serum was available only in no. 2016 and no. 2013, it is not clear whether the relatively high basal level of HA-specific serum IgG in no. 2016 is within the variability of ELISA measurement or due to some other reasons.” 

12) Please remove the last sentence on page 12, lines 195-196, and the first paragraph of page 13, lines 197-200, due to the fact that these pieces of information were already presented in the "introduction" section.

- We completely agree with the reviewer. We removed the first paragraph.

13) Although the suggestion presented on page 13, lines 209-211, are relevant, I believe that these pieces of information could be useful in to "Discussion" section. So, please remove it from the "Results" section to the "Discussion" section.

- Thank you for your suggestion. However, in many pieces of literature, the brief summary comment at the end of the section is helpful for readers to understand the contents and author’s claim. We prefer to leave the last sentence as it is.

14) In Fig. 3a it is possible to observe that the values of A(H1N1)pdm09-specific salivary IgA from the volunteer V11 were increased post-vaccination. Corroborating this observation the authors cited, on page 14, lines 217-218, that "...the IgA levels remained largely unchanged after vaccination except in V11." So, I would like to know whether the values obtained pré and post-vaccination were significantly different?

- We are sorry to confuse you. The values obtained pre and post-vaccination were not statistically different. We added the following sentence on page 15; Thus, basically, the sIgA was not increased by vaccination. 

- 

15) Are there statistical differences in the results presented in Figs. 4a and 4b?

- Thank you for the comment. The results are findings of serum and salivary antibodies only in three individuals. We think that the statistical analysis is not applicable here. 

16) Again, on page 14, lines 227-230, as well as on page 15, lines 242-247, the authors presented a putative conclusion or even suggestions for the results found. I recommend removing these sentences since this section must only present the results obtained in the study.

- We agree with the reviewer’s suggestion here. We removed these sentences to the discussion.

In the Discussion section:

17) The authors discuss, in a very well way, the potential use of saliva for monitoring not only the influenza virus infection but also its annual vaccination. However, some points could be improved:

 Although saliva could be considered a "corollary" fluid to evaluate the mucosal immunity of upper airways, there are some specific differences in oral and nasal mucosa immunity that must be discussed. Therefore, I suggest reorganizing some pieces of information from the "Introduction" section to the "Discussion" section.

 In addition, the "time" of mucosal immunity response should be more discussed since this fact could impact the evaluations presented in this study. For instance, as recently mentioned by Dos Santos et al.*, "...the literature demonstrates that SIgA is an important tool for early detection of infections, normally, 1 day after the infection, whereas serum IgA and IgM can be detected after 3–5 days after the infection (30)".

* Dos Santos JMB, Soares CP, Monteiro FR, Mello R, do Amaral JB, Aguiar AS, Soledade MP, Sucupira C, De Paulis M, Andrade JB, Almeida FJ, Sáfadi MAP, Mau LB, Brasil JM, Ramalho T, Loures FV, Vieira RP, Durigon EL, de Oliveira DBL, Bachi ALL. In Nasal Mucosal Secretions, Distinct IFN and IgA Responses Are Found in Severe and Mild SARS-CoV-2 Infection. Front Immunol. 2021 Feb 25;12:595343. doi: 10.3389/fimmu.2021.595343. eCollection 2021.

- Thank you for your helpful suggestions. We thoroughly re-organized the discussion. We hope you will find the resolution of your concerns in the new discussion section. 

- With respect to the time of mucosal response, the reference you have cited is the Dengue virus infection, not upper respiratory. Instead, I added the following in the discussion on page 20 (marked yellow); In this context, a wave of IgA plasma cells in the blood is considered to occur prior to the production of secretory IgA antibodies with a peak at around 6-10 days after mucosal infection/immunization (ref. No.24). As the reviewer #2 pointed out, the much earlier time course of antibody response in saliva would be interesting to be studied further in nasal vaccinations.

---

## [Decision Letter · Decision Letter 1]

19 Jan 2022

Saliva as a useful tool for evaluating upper mucosal antibody response to influenza

PONE-D-21-28281R1

Dear Dr. Tsunetsugu-Yokota,

We’re pleased to inform you that your manuscript has been judged scientifically suitable for publication and will be formally accepted for publication once it meets all outstanding technical requirements.

Kind regards,

Paulo Lee Ho, Ph.D.

Academic Editor

PLOS ONE

Additional Editor Comments (optional):

Reviewers' comments:

Reviewer's Responses to Questions

**Comments to the Author**

1. If the authors have adequately addressed your comments raised in a previous round of review and you feel that this manuscript is now acceptable for publication, you may indicate that here to bypass the “Comments to the Author” section, enter your conflict of interest statement in the “Confidential to Editor” section, and submit your "Accept" recommendation.

Reviewer #2: All comments have been addressed

2. Is the manuscript technically sound, and do the data support the conclusions?

Reviewer #2: Yes

3. Has the statistical analysis been performed appropriately and rigorously? 

Reviewer #2: Yes

4. Have the authors made all data underlying the findings in their manuscript fully available?

Reviewer #2: Yes

5. Is the manuscript presented in an intelligible fashion and written in standard English?

Reviewer #2: Yes

6. Review Comments to the Author

Reviewer #2: (No Response)

7. PLOS authors have the option to publish the peer review history of their article (what does this mean?). If published, this will include your full peer review and any attached files.

Reviewer #2: **Yes: **André LL Bachi

---

## [Editor Report · Acceptance letter]

28 Jan 2022

PONE-D-21-28281R1 

Saliva as a useful tool for evaluating upper mucosal antibody response to influenza 

Dear Dr. Tsunetsugu-Yokota:

I'm pleased to inform you that your manuscript has been deemed suitable for publication in PLOS ONE. Congratulations! Your manuscript is now with our production department. 

Kind regards, 

on behalf of

Dr. Paulo Lee Ho 

Academic Editor

PLOS ONE